Correspondence

EMBO
Molecular Medicine

# Amylopectinosis of the fatal epilepsy Lafora disease resists autophagic glycogen catabolism

Jun Wu [1], Or Kakhlon [2], Miguel Weil [3], Alexander Lossos[4] & Berge A Minassian [1]✉

Comment on: O Kakhlon et al (September 2021)

Misfolding causes a number of proteins to precipitate, aggregate and accumulate, with consequent neuroinflammation and neurodegeneration, e.g., in prion, Parkinson's, Pick's and Alzheimer's diseases. Misshaping of glycogen, specifically acquisition of overlong branches like those of plant starch amylopectin, causes it to precipitate, aggregate, and accumulate, with consequent neuroinflammation and neurodegeneration in adult polyglucosan body (APBD) and Lafora (LD) disease. Overlong-branched glycogen is termed polyglucosan, its aggregates polyglucosan bodies (PBs), and the process of polyglucosan formation and PB growth and spread across the brain amylopectinosis. Clinically, APBD resembles amyotrophic lateral sclerosis, while LD is a teenage-onset progressive myoclonus-epilepsy. In both, the PBs are diffusely present in neurons and astrocytes across the brain, with one difference in subcellular localization: neuronal PBs in APBD are mostly axonal, while in LD somatodendritic. APBD is caused by glycogen branching enzyme deficiency, and LD by deficiencies of the glycogen phosphatase laforin or its interacting E3 ubiquitin ligase malin. How laforin and malin regulate glycogen structure and how their deficiencies overextend glycogen branches are not known (Cenacchi et al, 2019).

LD is a disease with extreme severity. Within a few years from onset, the previously healthy child is continuously jerking in wakefulness (myoclonus), losing consciousness with each jerk, and hallucinating. This progressively worsens into a dementia, then vegetative state, and death in convulsive status epilepticus by 10 years from onset (Cenacchi et al, 2019).

The only known intracellular mammalian enzyme that can digest precipitated and aggregated polyglucosans is lysosomal acid maltase. Recently, Kakhlon and colleagues published in EMBO Mol Med extensive work leading to the development of the compound GHF201 (144DG11) that acts on the lysosomal membrane protein LAMP1 and increases autophagic flux. Applied to the APBD mouse model, this compound reduced PBs in the brain, heart, and liver, improved motor parameters, and dramatically improved mobility (see example video in original publication) and survival (60% at 449 days versus none in untreated mice) (Kakhlon et al, 2021). Recently (August 9, 2023), GHF201 received orphan designation for the treatment of APBD by the United States FDA (https://www.accessdata.fda.gov/scripts/opdlisting/oopd/detailedIndex.cfm?cfgridkey=953523). Three APBD patients are being treated with GHF201 on a compassionate basis under the care of co-author A. Lossos, the longest for almost 3 years now, and a formal clinical development program is in planning stages. The compound appears to be safe, and there appear to be objective improvements in motor strength, orthostatic blood pressure, and serum neurofilament light chain levels, among others. The potential applicability of the compound also to LD and the urgency of the need for a therapeutic for LD led us to test

GHF201 in the Epm2a⁻/⁻ laforin-deficient mouse model of the disease. The animal experiments were approved by the Institutional Animal Care and Use Committee at the University of Texas Southwestern Medical Center.

We used the same methods as were used in the APBD mouse model: 150 µl of 250 mg/kg in 5% DMSO injected subcutaneously twice a week from age 4 months until sacrifice at 10 months (Kakhlon et al, 2021). Tissue PB content was quantified histochemically (using HistoQuant software) and biochemically (as the insoluble fraction of total glycogen), as previously described (Nitschke et al, 2017; Sullivan et al, 2019).

We found no reduction in PBs in the brain as a whole, nor specifically in the hippocampus, motor cortex, piriform cortex, cerebellum, or striatum. There was also no reduction in the PB-accompanying astrogliosis and microgliosis. PBs were also not reduced in the two other major tissues that form PBs in LD mice, namely cardiac and skeletal muscle (Fig. 1).

Pathogenesis of polyglucosans in APBD is clear, glycogen branching insufficiency, but in LD remains opaque. Effectiveness of GHF201 on one and not the other adds a piece to the LD puzzle and the underlying unknown biology. Known ways in which LD polyglucosans differ from APBD polyglucosans are that on average the former: (1) have somewhat fewer overlong branches, (2) have somewhat larger volumes, and (3) are hyperphosphorylated (Sullivan et al, 2019). Remarkably, constitutively expressing

[1]Division of Neurology, Department of Pediatrics, University of Texas Southwestern Medical Center, Dallas, TX 75390, USA. [2]Department of Neurology, The Agnes Ginges Center for Human Neurogenetics, Hadassah Hebrew University Medical Center, Jerusalem 9112001, Israel. [3]Laboratory for Personalized Medicine and Neurodegenerative Diseases, The Shmunis School of Biomedicine and Cancer Research, The George S. Wise Faculty for Life Sciences, Sagol School of Neurosciences, Tel Aviv University, Tel Aviv, Israel. [4]Departments of Oncology and Neurology, Leslie and Michael Gaffin Center for Neuro-Oncology, Hadassah-Hebrew University Medical Center, Jerusalem, Israel. ✉E-mail: Berge.Minassian@utsouthwestern.edu
https://doi.org/10.1038/s44321-024-00063-9 | Published online: 2 April 2024

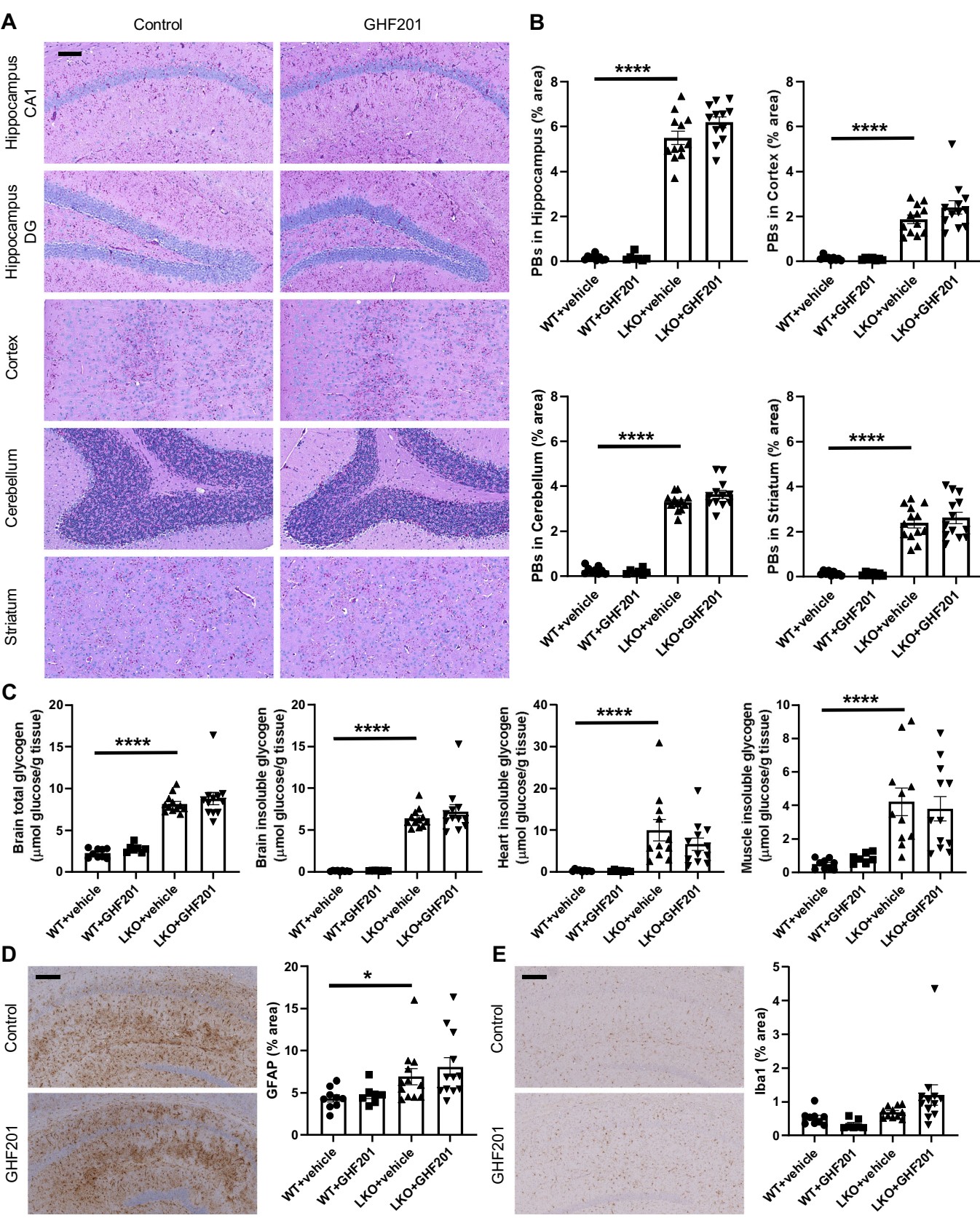

◄ **Figure 1. The autophagic activator GHF201 does not impact PBs in the *Epm2a⁻/⁻* mouse model of LD.**

(A) Representative images of PASD-stained hippocampus, cortex, cerebellum, and striatum of mice treated with vehicle control or GHF201. Scale bar: 100 μm. (B) Histochemical quantification of PBs in different brain regions. (C) Biochemical quantification of total and insoluble (PB) glycogen in brain, heart, and skeletal muscle. (D, E) Representative immunohistochemistry (IHC) images with signal quantification of anti-GFAP (D) and anti-Iba1 (E) in hippocampus showing no amelioration of astrogliosis or microgliosis, respectively. Scale bar: 200 μm. Data information: Quantification of (B–E) based on $n = 9$ WT vehicle-treated, $n = 7$ WT GHF201-treated, $n = 11$–12 LKO vehicle-treated, and $n = 12$ LKO GHF201-treated mice. Statistically significant differences (Mann-Whitney tests) were demonstrated between WT and LKO mice, but not between GHF201 and vehicle-treated LKO mice for (B–D). Significance levels are indicated as $*P < 0.05$; $****P < 0.0001$. All error bars represent SE. Source data are available online for this figure.

phosphatase-inactive but otherwise intact laforin in laforin-deficient mice prevents polyglucosan formation without correcting glycogen hyperphosphorylation, indicating that glycogen hyperphosphorylation is not causative of polyglucosan formation (Nitschke et al, 2017). But glycogen dephosphorylation must have a biological role since it is an unequivocal function of laforin (Tagliabracci et al, 2007; Worby et al, 2006). Perhaps glycogen dephosphorylation is required only when glycogen is precipitated (as polyglucosan), for it to be cleared by autophagy. This could explain why GHF201, which enhances autophagy, clears polyglucosans from the APBD but not the laforin-deficient LD mouse model, as we have shown here. Consistent with this, polyglucosans in malin-deficient LD are hyperphosphorylated like polyglucosans of laforin-deficient LD, even though the portion of glycogen that is properly shaped and soluble in malin-deficient LD is not hyperphosphorylated, unlike the same portion in laforin-deficient LD (Nitschke et al, 2017).

A second reason why LD polyglucosans resist clearance following activation of autophagy by GHF201 could be the presence in LD of a yet poorly understood primary defect in autophagy. Several groups presented evidence of this defect, although one could not corroborate it (Aguado et al, 2010; Criado et al, 2012; Puri et al, 2012; Wang et al, 2016). Possibly, two mechanisms could combine in LD to prevent constitutive or GHF201-enhanced autophagy from clearing PBs: defective marking for autophagy (by dephosphorylation) and a primary autophagic insufficiency.

Families of LD patients are desperate for a treatment, and the preliminary news on GHF201 in APBD is presently driving high expectations and requests to try the compound. The present work tempers these expectations, helps mitigate painful disappointment, and helps this particular rare disease community in their evaluations of where to invest hope and limited resources.

## Data availability

This study includes no data deposited in external repositories.

## Peer review information

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

## Acknowledgements

The work was supported by Chan Zuckerberg Initiative Patient-Partnered Collaborations for Rare Neurodegenerative Disease grant (2022-316703) and NIH P01NS097197. We acknowledge the Histopathology Core and Whole Brain Microscopy Facility (RR: SCR_017949) at University of Texas Southwestern Medical Center for performing histological staining and providing histological slide-scanning facilities.

## Author contributions

**Jun Wu**: Resources; Formal analysis; Investigation; Visualization; Methodology; Writing—review and editing. **Or Kakhlon**: Resources; Writing—review and editing. **Miguel Weil**: Writing—review and editing. **Alexander Lossos**: Writing—review and editing. **Berge A Minassian**: Conceptualization; Resources; Supervision; Funding acquisition; Investigation; Writing—original draft; Writing—review and editing.

## Disclosure and competing interests statement

Patent WO2018154578, awarded to OK and MW, pertains to GHF201 results. The authors declare no other competing interests.

