## [Peer Review File · EMBO Molecular Medicine]

Amylopectinosis of the fatal epilepsy Lafora disease resists autophagic glycogen catabolism

Jun Wu, Or Kakhlon, Miguel Weil, Alexander Lossos, and Berge Minassian

Corresponding author: Berge Minassian (Berge.Minassian@utsouthwestern.edu)

Review Timeline:

Submission Date:	18th Jan 24
Editorial Decision:	6th Mar 24
Revision Received:	14th Mar 24
Accepted:	19th Mar 24

Editor: Zeljko Durdevic

Transaction Report:

6th Mar 2024

Dear Prof. Minassian,

Thank you for the submission of your manuscript to EMBO Molecular Medicine and please accept my apologies for the delay in getting back to you. I am pleased to inform you that we will be able to accept your manuscript pending the following final amendments:

1) Author checklist: Please submit a complete checklist. <https://www.embopress.org/pb-assets/embosite/EMBO%20Press%20Author%20Checklist-1642513524327.xlsx>

2) In the main manuscript file, please do the following:

- Please address all comments suggested by our data editors listed below:

o Figure legends:

1. Please note that the figure is not labelled in the manuscript. This needs to be rectified.

2. Please note that the legends for figures e-f is not provided in the sequential manner (legend for figure 'f' is provided before legend of figure 'e'). This needs to be rectified.

3. Please define the annotated p values ****/* in the legend of figure b-c, e as appropriate.

4. Please indicate the statistical test used for data analysis in the legends of figures b-c, e.

5. Please note that information related to n is missing in the legends of figures b-c, e, g.

6. Please note that the error bars are not defined in the legends of figures b-c, e, g.

- Please include in the text information about ethics approval for the animal experiments.

- The figure should be renamed and called out as Figure 1. Please also call out individual panels in a sequential order.

- Please add "Disclosure Statement & Competing Interests". We updated our journal's competing interests policy in January 2022 and request authors to consider both actual and perceived competing interests. Please review the policy <https://www.embopress.org/competing-interests> and update your competing interests if necessary.

- Author contributions: CRediT has replaced the traditional author contributions section because it offers a systematic machine-readable author contributions format that allows for more effective research assessment. You are encouraged to use the free text boxes beneath each contributing author's name to add specific details on the author's contribution. More information is available in our guide to authors:

<https://www.embopress.org/page/journal/17574684/authorguide#authorshipguidelines>

- Please provide data availability statement. If no data were deposited add the sentence "This study includes no data deposited in external repositories".

3) Acknowledgments: Please make sure that information about all sources of funding are complete in both our submission system and in the manuscript.

4) Please include one (two) sentence summary of your findings in the point-by-point response.

5) As part of the EMBO Publications transparent editorial process initiative (see our Editorial at <http://embomolmed.embopress.org/content/2/9/329>), EMBO Molecular Medicine will publish online a Review Process File (RPF) to accompany accepted manuscripts. This file will be published in conjunction with your paper and will include the anonymous referee reports, your point-by-point response and all pertinent correspondence relating to the manuscript. Let us know whether you agree with the publication of the RPF and as here, if you want to remove or not any figures from it prior to publication. Please note that the Authors checklist will be published at the end of the RPF.

6) Please provide a point-by-point letter INCLUDING my comments as well as the reviewer's reports and your detailed responses (as Word file).

I look forward to reading a new revised version of your manuscript as soon as possible.

Yours sincerely,

Zeljko Durdevic

*** Instructions to submit your revised manuscript ***

*** PLEASE NOTE *** As part of the EMBO Publications transparent editorial process initiative (see our Editorial at

<https://www.embopress.org/doi/pdf/10.1002/emmm.201000094>), EMBO Molecular Medicine will publish online a Review Process File to accompany accepted manuscripts.

1) a .docx formatted version of the manuscript text (including Figure legends and tables)

2) Separate figure files*

3) supplemental information as Expanded View and/or Appendix. Please carefully check the authors guidelines for formatting Expanded view and Appendix figures and tables at <https://www.embopress.org/page/journal/17574684/authorguide#expandedview>

4) a letter INCLUDING the reviewer's reports and your detailed responses to their comments (as Word file).

5) The paper explained: EMBO Molecular Medicine articles are accompanied by a summary of the articles to emphasize the major findings in the paper and their medical implications for the non-specialist reader. Please provide a draft summary of your article highlighting

This may be edited to ensure that readers understand the significance and context of the research.

Please refer to any of our published articles for an example.

6) For more information: There is space at the end of each article to list relevant web links for further consultation by our readers. Could you identify some relevant ones and provide such information as well? Some examples are patient associations, relevant databases, OMIM/proteins/genes links, author's websites, etc...

7) Author contributions: the contribution of every author must be detailed in a separate section.

8) EMBO Molecular Medicine now requires a complete author checklist (<https://www.embopress.org/page/journal/17574684/authorguide>) to be submitted with all revised manuscripts. Please use the checklist as guideline for the sort of information we need WITHIN the manuscript. The checklist should only be filled with page numbers where the information can be found. This is particularly important for animal reporting, antibody dilutions (missing) and exact values and n that should be indicated instead of a range.

9) Every published paper now includes a 'Synopsis' to further enhance discoverability. Synopses are displayed on the journal webpage and are freely accessible to all readers. They include a short stand first (maximum of 300 characters, including space) as well as 2-5 one sentence bullet points that summarise the paper. Please write the bullet points to summarise the key NEW findings. They should be designed to be complementary to the abstract - i.e. not repeat the same text. We encourage inclusion of key acronyms and quantitative information (maximum of 30 words / bullet point). Please use the passive voice. Please attach these in a separate file or send them by email, we will incorporate them accordingly.

You are also welcome to suggest a striking image or visual abstract to illustrate your article. If you do please provide a jpeg file 550 px-wide x 300-800px high.

10) A Conflict of Interest statement should be provided in the main text

11) Please note that we now mandate that all corresponding authors list an ORCID digital identifier. This takes <90 seconds to complete. We encourage all authors to supply an ORCID identifier, which will be linked to their name for unambiguous name identification.

Currently, our records indicate that the ORCID for your account is 0000-0002-9322-0189.

Please click the link below to modify this ORCID:
Link Not Available

Photos 400-800 DPI

*Additional important information regarding figures and illustrations can be found at
<https://bit.ly/EMBOPressFigurePreparationGuideline>. See also figure legend preparation guidelines:
<https://www.embopress.org/page/journal/17574684/authorguide#figureformat>

***** Reviewer's comments *****

Referee #2 (Novelty/Model system Comments for Author):

The authors use a well-established model of Lafora disease. The statistical analysis is appropriate. They apply a previously described approach to test GHF201 in the APBD mouse model. GHF201 has shown benefit in the APBD mouse model and perhaps in some humans with APBD.

The authors clearly elaborate on why GHF201 would have been hoped to benefit LD. They find no improvement in their LD model suggesting it may not be efficacious in humans with LD. The authors only describe immunohistochemical assessments and no details about motor function or survival. This could have added to the story but is not absolutely necessary.

While I agree that these results strongly temper enthusiasm for the use of GHF201 in humans with LD, species differences are not uncommon in therapeutic trials. If GHF201 has a favorable side effect profile, it may be not unreasonable to try GHF201 in humans with LD even with these results.

Referee #2 (Remarks for Author):

The article is well written and clear. The science is sound and impactful. I thank the authors for this good work.

***** Reviewer's comments *****

We appreciate the reviewer for taking time to carefully review the manuscript and give constructive comments. Below is our point-by-point response to each comment.

Referee #2 (Novelty/Model system Comments for Author):

The authors use a well-established model of Lafora disease. The statistical analysis is appropriate. They apply a previously described approach to test GHF201 in the APBD mouse model. GHF201 has shown benefit in the APBD mouse model and perhaps in some humans with APBD.

The authors clearly elaborate on why GHF201 would have been hoped to benefit LD. They find no improvement in their LD model suggesting it may not be efficacious in humans with LD. The authors only describe immunohistochemical assessments and no details about motor function or survival. This could have added to the story but is not absolutely necessary.

Since no significant difference in motor function or survival could be observed between wildtype mouse and LD mouse model in Minassian lab of UTSouthwestern Medical Center, no assessment of behavior is shown in the present study.

While I agree that these results strongly temper enthusiasm for the use of GHF201 in humans with LD, species differences are not uncommon in therapeutic trials. If GHF201 has a favorable side effect profile, it may be not unreasonable to try GHF201 in humans with LD even with these results.

That is a good point, and the drug could be tried 'off-label' on Lafora patients, as it further proves safe in APBD, and perhaps starts exhibiting efficacy signal.

Referee #2 (Remarks for Author):

The article is well written and clear. The science is sound and impactful. I thank the authors for this good work.

19th Mar 2024

Dear Prof. Minassian,

We are pleased to inform you that your manuscript is accepted for publication and is now being sent to our publisher to be included in the next available issue of EMBO Molecular Medicine.
